# Metabolomic Profiling of Adults with Congenital Heart Disease

**DOI:** 10.3390/metabo11080525

**Published:** 2021-08-09

**Authors:** Ari Cedars, Cedric Manlhiot, Jong-Mi Ko, Teodoro Bottiglieri, Erland Arning, Angela Weingarten, Alexander Opotowsky, Shelby Kutty

**Affiliations:** 1Department of Pediatrics, Johns Hopkins University, Baltimore, MD 21218, USA; cmanlhi1@jhmi.edu (C.M.); jko24@jhmi.edu (J.-M.K.); skutty1@jhmi.edu (S.K.); 2Center of Metabolomics, Baylor Scott & White Research Institute, Dallas, TX 75204, USA; teodoro.bottiglieri@bswhealth.org (T.B.); erland.arning@bswhealth.org (E.A.); 3Department of Medicine, Vanderbilt University, Nashville, TN 37235, USA; Angela.j.Weingarten@vumc.org; 4Department of Cardiology, Cincinnati Children’s Hospital, Cincinnati, OH 45229, USA; sasha.opotowsky@cchmc.org

**Keywords:** adult congenital heart disease, metabolomic analysis, biomarkers

## Abstract

Metabolomic analysis may provide an integrated assessment in genetically and pathologically heterogeneous populations. We used metabolomic analysis to gain mechanistic insight into the small and diverse population of adults with congenital heart disease (ACHD). Consecutive ACHD patients seen at a single institution were enrolled. Clinical variables and whole blood were collected at regular clinical visits. Stored plasma samples were analyzed for the concentrations of 674 metabolites and metabolic markers using mass spectrometry with internal standards. These samples were compared to 28 simultaneously assessed healthy non-ACHD controls. Principal component analysis and multivariable regression modeling were used to identify metabolites associated with clinical outcomes in ACHD. Plasma from ACHD and healthy control patients differed in the concentrations of multiple metabolites. Differences between control and ACHD were greater in number and in degree than those between ACHD anatomic groups. A metabolite cluster containing amino acids and metabolites of amino acids correlated with negative clinical outcomes across all anatomic groups. Metabolites in the arginine metabolic pathway, betaine, dehydroepiandrosterone, cystine, 1-methylhistidine, serotonin and bile acids were associated with specific clinical outcomes. Metabolic markers of disease may both be useful as biomarkers for disease activity and suggest etiologically related pathways as possible targets for disease-modifying intervention.

## 1. Introduction

The field of metabolomics has the potential to dramatically impact research and progress in cardiology. As metabolomic changes represent the integrated effects of genomic, proteomic and environmental or dietary influences, they may provide unique insights into disease pathobiology [1]. By providing information on the substrates participating in a broad diversity of biological processes, metabolomic analysis can indicate changes in these processes which may be contributing to disease. The metabolome is thus proposed to be the closest readout of disease status in an individual at any given time [2]. In the field of acquired heart disease, metabolomic studies have already led to significant new insights, revealing previously unrecognized biomarkers [3] and pathophysiological mechanisms [4,5,6]. Similarly in the field of congenital heart disease, metabolomic studies have revealed informative changes in the plasma concentrations of amino acids related to the nitric oxide signaling pathway [7] and in Fontan patients have revealed alterations in the concentrations of amino acids and lipid metabolites [8]. Existing studies are limited, however, in both the numbers of metabolites and numbers of patients investigated. In fact, there is a comparative paucity of metabolomic data related to cardiac disease generally, in part as a result of technical and practical challenges [9], making metabolomics a largely untapped resource for biological information.

Metabolomic analysis may offer unique insights into the pathology underlying clinical deterioration in adult congenital heart disease (ACHD). In spite of growing prevalence [10] and high rates of hospitalization [11] and mortality [12] there remain few interventions with proven efficacy in preventing clinical deterioration in ACHD [13]. In part this failure is attributable to prohibitively complex genetic, anatomical, physiologic and environmental systemic inputs in this group of patients which is too small and heterogeneous to permit population-based research. In this scenario, the integrative capacity of metabolomic analysis has the potential to uncover unifying and potentially targetable metabolic phenotypes generated by a broad diversity of upstream systemic inputs. In support of this hypothesis, the pairing of metabolomic analysis with computer modeling has recently yielded novel insight into the etiologically heterogeneous pathophysiology of acquired systolic heart failure [14].

In the present study, we investigated the potential utility of metabolomic analysis to generate new hypotheses and provide new insights into the pathophysiology of ACHD.

## 2. Results

### 2.1. Study Cohort

The final cohort included a total of 167 assessment/biological sample pairs, representing 99 individual patients (61 individuals with a single sample, 18 with 2, 14 with 3 and 6 with 4+ samples). Fifty-one percent were male and 76% Caucasian with an average age at assessment of 38.3 ± 13.3 years old. ACHD diagnoses were 21% SRV, 20% SD, 35% RVOT, 9% SV and 15% other. There were 26 non-ACHD healthy control samples having an average age of 43.7+7.7 years old (older than the ACHD cohort *p* = 0.008) with 42% males (*p* = 0.51 compared to the study cohort). Cohort characteristics can be found in Table 1 and patient’s clinical status at the time of study evaluation is detailed in Appendix A. The non-cardiac diagnoses present in each of the 71 individuals who had them (excluding those listed in Table 1) can be found in Appendix A.

### 2.2. Metabolite Differences between ACHD and Control

Amino acids and related metabolic compounds which differed significantly between ACHD and controls are depicted in Figure 1 while a comparison of lipid-derived metabolic compounds are reported in Figure 2. The degree of variation between ACHD patients and controls for all metabolites are depicted in Volcano plots, Appendix A. Overall, plasma concentrations differed for a large number of amino acids and related metabolic compounds between ACHD and controls with no striking differences between the various ACHD groups. We found lower concentrations of serotonin, serotonin/tryptophan ratio and hypoxanthine across all ACHD diagnosis groups compared to control. Patients with RVOT, SRV and SD all had lower plasma concentrations of lactate and higher concentrations of arginine than those observed in control samples. There were no significant differences between ACHD and controls in serum concentrations of lipids lumped by class (as depicted in Figure 2) or in specific individual lipid molecular species.

All metabolites with uncorrected *p*-values < 0.05 for a difference between ACHD and control were investigated further by metabolic pathway analysis. Table 2 depicts significantly impacted pathways (*p* < 0.05) along with the level of significance corrected for a false discovery rate of 0.05 with level of significance as indicated.

### 2.3. ACHD Principal Component Cluster Determination

In principal component analysis, the first four components individually were retained for further analysis based on an examination of the scree plot. These components respectively explained 30%, 9%, 5% and 4% of the variance and together explained 48% of cumulative variance in metabolic compound concentrations, Appendix A. Investigating these components more closely revealed that the first principal component (PC1) almost exclusively includes metabolites from the triacylglycerol class, and the second principal component (PC2) includes metabolites from the sphingolipid, glycerophospholipid, ceramide and cholesterol ester classes. The third principal component (PC3) is dominated by acylcarnitines and glycerophospholipids. The fourth principal component (PC4) included many amino acids and related compounds. Table 3 depicts the metabolic pathways represented in the various principal components.

### 2.4. Metabolite Association with Clinical Variables

Multivariable regression models were constructed as described in the methods section and included the first four principal components along with sex, race, age, diagnosis, presence of non-cardiac diagnoses, hypertension, diabetes and dyslipidemia as potential covariates. As depicted in Table 4 (left panel), a lower patient score on PC1 was associated with increased odds of arrhythmia history and shortness of breath. A higher patient score on PC2 was associated with increased odds of arrhythmia history. A lower patient score on PC4 was associated with an increased probability of all adverse cardiac characteristics and variables (both measured and self-reported) and worse self-reported quality of life Figure 3.

Volcano plots were created for each adverse cardiac variable in an effort to better understand which specific metabolites were most strongly associated with patient characteristics and outcomes, Appendix A. Most notable are arginine pathway metabolites (arginine, methylarginines and citrulline), 1-methylhistidine, ornithine and dehydroepiandrosterone. The pathways involved with these metabolites are reported in Appendix A.

Finally, we compared regression models generated by combining the metabolites with high reliability identified in the bootstrap analysis for each clinical variable, Appendix A, and principal component loads with those generated using principal component loads alone. The results are depicted on the right side of Table 4. Regression models generated by adding the individual metabolites with high reliability showed better fit (lower QIC) for all clinical characteristics and variables with the exception of exertional shortness of breath. Importantly, this analysis did not find any individual metabolite to be associated with the broad diversity of clinical characteristics and variables as was found for PC4.

## 3. Discussion

In the present analysis, we investigated plasma metabolite levels in an anatomically heterogeneous outpatient ACHD cohort. We found significant differences in plasma metabolite concentrations both between ACHD and non-ACHD patients, and between ACHD patients with differing physiologic abnormalities. We also found that metabolic phenotype did not necessarily align with physiologic abnormality; but correlated well with cardiac function, oxygen saturation and patient-reported health status. We specifically found that a cluster of metabolites including intermediates in amino acid metabolic pathways had a strong positive relationship with cardiac outcomes and health status. Analysis of the specific metabolites involved suggests metabolites in the arginine, choline, steroid hormone and bile acid pathways as potential biomarkers and targets for further therapeutic investigation. While we did not identify any biomarker unique to ACHD (for which non-targeted metabolomics studies would likely be better suited) we did identify metabolic shifts shared with other forms of chronic heart disease.

In the present study, we chose to perform a targeted metabolomics approach inclusive of a broad array of metabolites. We selected this approach given the availability of excellent existing untargeted metabolomic analyses in heart failure and coronary vascular disease [3,5,15,16,17]. As the most common causes for mortality in ACHD are heart failure and arrhythmia [18,19], similar to what is seen in acquired heart disease, we anticipated that changes in recognized, informative plasma metabolite levels might be similarly informative in ACHD. We also were interested in being able to specifically quantify and compare the relative abundance of metabolites in the same pathway as potentially useful in identifying pathway flux, for which untargeted analysis is more limited.

Using this approach we identified high probability metabolic markers of cardiac function and clinical status. Among the metabolic pathways we identified as potentially informative, arginine metabolism is particularly interesting. Metabolites in this pathway segregated to metabolic cluster 4 (which correlated with cardiac function and clinical status), were independently associated with these outcomes and were significantly different when comparing ACHD and non-ACHD. One of the fundamental roles played by the arginine metabolic pathway is the elaboration of the biologically active signaling molecule nitric oxide. Alterations in the concentrations of metabolite members of this pathway correlate with prognosis in various types of heart disease [5,20,21] including congenital heart disease generally [7]. Moreover, one prior study demonstrates that ADMA is elevated in Fontan patients compared to controls although no association with clinical status or outcome was identified [22]. In the present analysis we identified alterations specifically in the concentrations of asymmetric dimethyl arginine and symmetric dimethyl arginine which may impact arginine transport and nitric oxide synthesis [21,23]. We also identified evidence of altered flux through the nitric oxide pathway with increased total ornithine, ornithine to arginine ratio and total dimethyl arginine to arginine ratio. The overall metabolite pattern identified suggests the hypothesis that elevated levels of dimethylarginine in clinically deteriorating ACHD patients may lead to arginine diversion away from nitric oxide synthesis, potentially adversely impacting clinical status and leading to disease progression in ACHD as in other diseases. Given the availability of nitric oxide signaling pathway modulators for use in pulmonary hypertension, this metabolic pathway has potential as both a source for informative biomarkers and as a possible therapeutic target.

We also found that increased levels of the quaternary amine betaine correlate with hypoxemia. Betaine is a member of the choline metabolic pathway. Previous studies suggest that alterations in serum levels of members of the choline metabolic pathway, including betaine, are associated with risk for cardiovascular disease [4,6]. These studies indicate that alterations in diet may impact choline synthesis at the level of the gut microbiome. How hypoxemia or conditions associated with hypoxemia (which in the present study included mostly SV patients who have chronic gut congestion) might impact the gut microbiome and choline biosynthesis may prove an avenue for further research.

We identified an informative association between decreased dehydroepiandrosterone (DHEA) levels and RV dysfunction. DHEA has been demonstrated in animal models to reverse pulmonary vascular disease and improve RV function in the setting of pulmonary hypertension [24,25,26]. Moreover, DHEA has been directly associated with RV systolic function in humans in a large population-based cohort [27]. This finding suggests the possibility that either RV dysfunction or the sequelae of RV dysfunction (such as central venous congestion) lead to DHEA elaboration. The association between lower DHEA and RV dysfunction suggests the hypothesis that breakdown of a homeostatic signaling mechanism involving DHEA may contribute to propagation of RV dysfunction in ACHD. If confirmed, this will be an important avenue for further mechanistic investigation.

We also identified other potentially informative associations between metabolites and clinical status in ACHD. These included alterations in the levels of bile acids, specifically glycochenodeoxycholic acid and taurocholic acid which have been demonstrated to be elevated in the setting of liver injury [28] and in our analysis were associated with SV patients who are known to have chronic hepatic congestion. Elevated levels of 1-methylhistidine in our study correlated with general cardiac dysfunction, which is consistent with existing data showing plasma concentrations of the modified amino acid correlate with both systolic and diastolic cardiac dysfunction in heart failure patients [29,30]. In contrast the observed decreased plasma concentrations of reduced cysteine (cysteine) and serotonin/tryptophan do not have a clear explanation in existing literature. If validated in larger cohorts, these metabolites are worth pursuing as potentially informative biomarkers in the future.

While we identified multiple individual informative metabolites, none of these metabolites was consistently associated with outcomes across clinical groups. Many of these metabolites, however, are components of metabolic cluster 4 which was associated with adverse clinical outcomes across groups. This suggests that multiple metabolic pathways, dysfunctionally regulated by a diverse array of upstream stimuli, may act in concert to produce or reflect clinical disease in the ACHD population. In possible support of this hypothesis, we found that metabolic cluster 4 and not individual metabolites was associated with a better fit for the symptom of exertional shortness of breath. Given that physical exertion produces a stress on the entire hemodynamic system and is a non-specific symptom of disease, this finding may suggest that multiple metabolic systems, as defined in this metabolic cluster, are etiologically related to disease progression. It also suggests that no single pathway in isolation holds the key to understanding and preventing clinical deterioration in ACHD. Looking forward this may indicate that a multi-intervention approach will be required to effectively manage the sequelae of ACHD, not dissimilar to what is the case in systolic heart failure [31].

### Limitations

The present data and analyses are intended to be hypothesis generating. Samples were drawn without regard to fasting status or time of day for the sake of practicality (samples were obtained at the time of regular clinical visits). Given known impact of both diet and diurnal variation in hormone levels on serum metabolite concentrations, this may have an impact on the results. Nevertheless, recent data suggest the impact of fasting, activity and time of day have limited impact on plasma metabolite profile [32]. Moreover we employed a targeted metabolomics approach in the present analysis which limits analysis to only the restricted set of quantified metabolites. As a result, informative metabolites not specifically investigated in our metabolite panel would be missed by our approach. Relevant to this consideration, given the very broad diversity of lipid metabolites quantified, we elected not to compare abundance of specific lipid species between the ACHD and control groups as part of the present analysis, instead reserving this for a future analysis. There is a significant possibility that sampling error may have led to false associations given the limited sample size employed in the present analysis. These findings will require validation in larger more homogeneous cohorts. This study analysis was retrospective. As such, not all patients had all assessments at all encounters resulting in some outcomes having a lower effective sample size. Moreover, although we controlled for all factors possible, metabolomic analysis is inherently sensitive to diet and physical activity which were not controlled for in the present analysis. Similarly, it is possible that there was some systematic error introduced into the analytic framework at the level of sample analysis. We have tried to identify these system biases as possible. Finally, certain of the outcomes were patient reported and therefore subject to bias based on psychological state at the time of assessment.

## 4. Materials and Methods

We conducted a retrospective study including patients enrolled in an ACHD clinical database and biobank at Baylor University Medical Center. This study was approved by the institutional review boards at Baylor University Medical Center and the University of Texas Southwestern Medical Center and was conducted in accordance with the Helsinki declaration and the International Conference on Harmonization Good Clinical Practice (ICH-GCP) guidelines. Each participant provided signed informed consent to participate in the study. All biological materials and clinical data will be made available to qualified investigators upon reasonable request.

### 4.1. Study Cohort

Patients meeting inclusion criteria and seen in the outpatient ACHD clinic at the enrolling site were approached consecutively and offered enrollment by the primary investigator at the time of regular outpatient visits. The inclusion criteria included age ≥ 18 years, a diagnosis of ACHD confirmed by the principal investigator, being regularly followed in the ACHD clinic, mental capability and ability to provide informed consent. Blood samples were not collected during pregnancy. Blood samples and clinical data were assessed serially at each clinically indicated visit such that some patients had multiple samples and multiple clinical assessments. Due to heterogeneous periods between visits, there was no defined time between repeated samples. Twenty-six control samples were identified from among previously collected healthy volunteer samples who were free of chronic disease and took no medications.

### 4.2. Clinical Data Acquisition

Clinical data were obtained from the enrolling site clinical database and biobank. This database includes both data abstracted from the electronic health record for each patient and clinical status directly assessed at the time of each visit. This dataset included the following variables: Demographics: Age, race, ethnicity, gender, years of education, highest degree, occupation annual income; health history: Tobacco use, pregnancy history, arrhythmia history and type, arrhythmia ablation, pacemaker type, implantable cardiac defibrillator, ACHD lesion type, cardiac surgery type, vascular surgery type, noncardiac surgery type, noncardiac diagnosis type, Eisenmenger syndrome, hypertension, diabetes mellitus, dyslipidemia, renal disease, dialysis, stroke, peripheral vascular disease; medications: Medication identity, dose, indication, frequency, start and stop dates; family health history; cardiac function: Right ventricular systolic function, left ventricular systolic function, valvular function (for all cardiac functional assessments mode of assessment and date were obtained); clinical characteristics and outcomes: Hospitalization including discharge diagnosis, transplantation, ventricular assist device implant, death; cardiac functional parameters: > moderate left ventricular systolic dysfunction (LV ejection Fraction < 40%), > moderate right ventricular systolic dysfunction, > moderate valvular dysfunction (stenosis or regurgitation); vital signs: Blood pressure, heart rate, oxygen saturation on room air or on supplemental oxygen; ACHD Patient Reported Outcome (PRO) score: ACHD disease-specific health status questionnaire. Physiologically similar patients were then grouped together to ensure adequate feature counts for analysis. These groups included: Systemic right ventricle (SRV), right ventricular outflow tract (RVOT) lesions including Tetralogy of Fallot (TOF) and double outlet right ventricle (DORV; excluding Taussig–Bing anomaly), septal defects (SD), single ventricle/Ebstein’s anomaly (SV) and other (left ventricular outflow tract stenosis, hypertrophic cardiomyopathy, aortopathy or coronary anomalies). All data were collected in a REDCap database. Any variable with > 50% missingness was excluded from further analysis. Patient characteristics and clinical profiles are reported as means with standard deviation, median with interquartile range and frequencies as appropriate.

### 4.3. Biological Sample Acquisition and Metabolite Quantification

From participants who consented to contribute blood samples for research, about 30 mL of whole blood was collected in serologic vacuum evacuated tubes (1 EDTA (10 mL) and 2 lithium-heparin (20 mL)) at the initial and follow-up visits. Each follow-up visit took place at different time point in ACHD clinic as part of routine clinical care or at the time of admission to the hospital for heart-related invasive or surgical procedures. A blood draw was not performed on participants who refused at any time or who had become pregnant. Within 1 h of sample collection, tubes were spun in a hematologic centrifuge for 15 min to separate plasma from blood cells. Plasma was then aspirated from the cells using a transfer pipette and divided into aliquots in labeled 2 ml screw-top freezer tubes. To ensure de-identification of samples, an alphanumerical code was assigned to each patient’s sample. Samples were stored immediately after aliquoting in an onsite −20 degree Celsius freezer and then transferred weekly to a −80 degree Celsius freezer in the Center for Metabolomics at Baylor Scott and White Research Institute, Dallas, TX, for longer term storage.

Targeted metabolomic analysis was performed using the MxP^®^ Quant 500 metabolomic kit (Biocrates Life, Sciences AG, Innsbruck, Austria) according to the manufacture’s protocol for a 5500 QTrap (Sciex Framingham, MA, USA) mass spectrometer equipped with an electrospray ionization source using liquid chromatography tandem mass spectrometry (LC-MS/MS) and flow-injection analysis tandem mass spectrometry (FIA-MS/MS). This commercially available platform can detect and quantify 674 metabolites and metabolic indicators that cover 23 different classes of compounds. Pre-processing and analysis of plasma samples was performed in a 96 well plate supplied with the kit which contained calibration standards and spiked human quality control plasma at 3 different levels. Identification of compound mass spectra was performed using Analyst 6.0 (Sciex, Framingham, MA, USA). Data was exported and uploaded to MetIDQ software (Biocrates Life Sciences AG, Austria) for quantification from internal standards and calibration curves. The entire set of samples for this study required 4 plates and data was normalized across each plate using the median of QC level 2 (4 replicates/plate × 4 plates). In addition, we used an add-on software tool, MetaboINDICATOR™ (Biocrates Life Sciences AG, Austria), to calculate metabolite sums and ratios that are associated with specific pathways. This analysis potentially provides 232 metabolic indicators.

### 4.4. Metabolic Data Preprocessing

Metabolites that were below the limit of detection were replaced by the lower limit of detection for the assay divided by 2. Correction for batch effect in the measurement of metabolite concentration was done using the ComBat package on R.

### 4.5. Statistical Analysis—Sample Selection

In order to build the study cohort, we first identified available samples and paired them to clinic visits. Given that there were more clinical evaluations than samples available, a greedy matching algorithm without replacement was used to match available samples to a clinical evaluation within a 1-month period; this threshold was selected because it maximized the number of samples available for analysis while minimizing the time lag between sample collection and acquisition of clinical data. Some patients had > 1 sample available for analysis, in those cases, the earliest available pair of clinical assessment/sample for each patient (to prevent variations specific to an individual patient from being over-represented), was used to compare metabolite concentration between the various ACHD diagnostic groups and controls, all other analyses included all available samples with proper statistical adjustments for repeated measures in a single patient (described below).

### 4.6. Comparisons of Metabolic Profiles and Identification of Metabolic Pathways

Comparison of metabolite levels between ACHD and control, between the various ACHD groups and between ACHD patients with and without outcomes of interest was accomplished by calculating the fold difference between groups for each metabolite and statistical significance using Student’s t-test. These results were used to construct representative Volcano plots using the EnhancedVolcano package on R. To facilitate metabolite comparisons between the various diagnostic groups we calculated the normalized difference in metabolite concentrations between the various types of ACHD and healthy control patients along with 95% confidence limits and reported those graphically. Specific metabolites selected for this analysis included all metabolites for which the difference between ACHD patients and control or between at least one of the diagnostic groups and control was statistically significant using a false discovery rate of 5% (Benjamini-Hochberg method). For metabolic pathway analyses, we included all metabolites that were differentially expressed with a *p*-value of 0.05 or less and for which an HMDB ID was available. Pathway analyses were performed using the MetaboAnalyst tool [33] using the hypergeometric test for over-representation analysis and relative-betweeness centrality for pathway topology analysis. The October 2019 version of the KEGG was used as the pathway library. All impacted pathways (*p*-value < 0.05) identified in those analyses are reported along with the associated false discovery rate threshold.

### 4.7. Principal Component Analysis

In order to determine the association between metabolite concentration, patient characteristics and outcomes, principal component analysis was performed on all metabolites in the ACHD patient group. First, metabolite concentrations were normalized (scaled and centered). Based on an examination of the scree plot of the principal component loads, the first 4 components (respectively explaining 30%, 9%, 5% and 4% of the variance) were retained for further analysis. Metabolic pathway profiling for each principal component was conducted using the top 50 contributing metabolites in each principal component.

### 4.8. Association between Clinical Variables and Metabolic Profiles

We next investigated the association between metabolite concentrations and an array of clinical variables including: O2 saturation <95%, history of arrhythmia, general cardiac dysfunction (defined as LV ejection Fraction <40%, moderate or greater right ventricular systolic dysfunction or at least moderate valvular dysfunction of any valve), specific right ventricular (RV) dysfunction (defined as moderate or greater RV systolic dysfunction or moderate or greater tricuspid or pulmonary valve dysfunction), baseline shortness of breath, exertional shortness of breath, an ACHD disease specific patient-reported outcome metric (summary score and scores in each domain as previously published [34] and self-reported health-related quality of life (based on a single item patient-reported outcome question).

In order to investigate the association between the different principal components and clinical variables, we used linear (continuous variables) and logistic (binary variables) regression models, adjusted for repeated measures through either an independent or exchangeable (based on the model QIC) covariance structure as appropriate. All models were built in a stepwise fashion (*p* < 0.05 to enter/remain). Potential factors included all principal components scores along with the following clinical covariates: Sex, race, age, diagnosis, presence of a non-cardiac diagnoses, hypertension, diabetes, dyslipidemia and medications. A full list of patient medications for each patient will be made available upon request. Mean imputation was used for missing clinical covariates. A second version of these regression models was created in which the top metabolic compounds for each clinical variable, as determined based on statistical significance (*p* < 0.01) and fold difference (FD: 0.80/1.25) were included as candidate features. In order to facilitate feature selection we used a bootstrap resampling approach to first identify features with high reliability. In short, bootstrap resampling is a strategy where a number of subsamples (in this case 500 samples including between 20 and 80% of the total observations) are randomly drawn from the original cohort and a regression model is fitted to the specific subsample. The percentage of subsamples in which a specific feature is included in the final multivariable model for a given subsample is called the reliability score and features with scores >50% are considered highly reliable. For this analysis, only highly reliable features were potential candidates for inclusion in the regression models, Appendix A. Model features and building strategies were otherwise similar to the models including only the principal components and clinical covariates as potential features.

All analyses were performed using R v3.6.1 [35] with the stats and mlbench packages along with the other packages mentioned above and SAS (Cary, NC, USA) v9.4 using standard procedures. Figures were created using GraphPad 6 (San Diego, CA, USA).

## 5. Conclusions

We have identified metabolomic differences between ACHD and non-ACHD individuals, and have described metabolic variations in the ACHD population which correlate with cardiac function and health status. We anticipate that these data will lead to new avenues for research in the field with the goal of identifying the mechanisms underlying disease progression in ACHD and informative biomarkers to guide therapeutic intervention.

## Figures and Tables

**Figure 1 metabolites-11-00525-f001:**
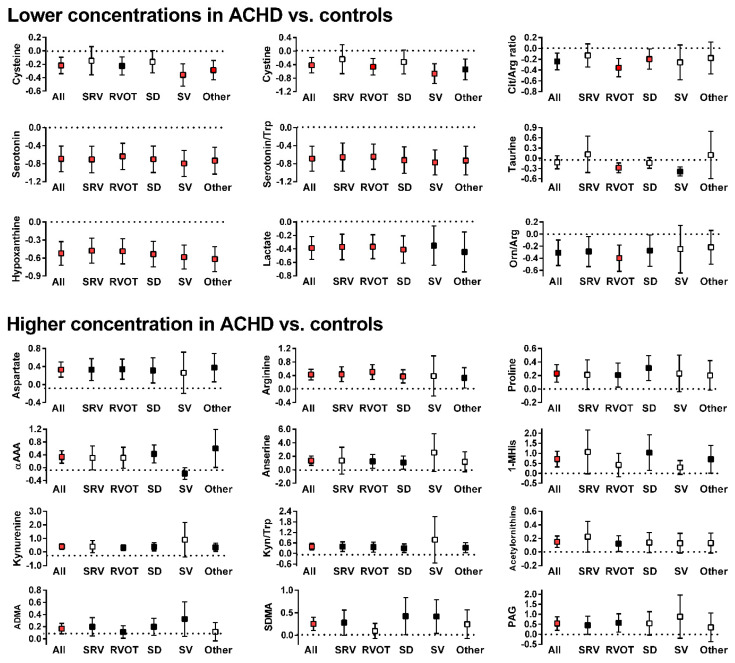
Significant differences in concentration of amino acids and related metabolic compounds between patients in any of the ACHD groups and controls (box represent the normalized differences along with 95% confidence interval; dotted line at x = 0 is the concentration in control subjects. White boxes: Difference is not statistically significant; black boxes: Difference is statistically significant but is above the threshold for a false discovery rate of 5%; red boxes: Difference is statistically significant and is below the threshold for a false discovery rate of 5%). Y-axis values indicate fold concentration difference between ACHD and controls for the indicated variable. x-axis abbreviations: ALL—all diagnoses; OTH—other diagnoses; SRV—transposition of the great arteries; RVOT—Tetralogy of Fallot/right ventricular outflow stenosis; SD—septal defect; SV—single ventricle. Metabolites abbreviations: 1-MHis—1-Methylhistidine; αAAA—aminoadipic acid; ADMA—asymmetric dimethylarginine; Arg—arginine; Cit-citrulline; Kyn—kynurenine; Orn—ornithine; PAG—phenylacetylglycine; SDMA—symmetric dimethylarginine; Trp—tryptophan.

**Figure 2 metabolites-11-00525-f002:**
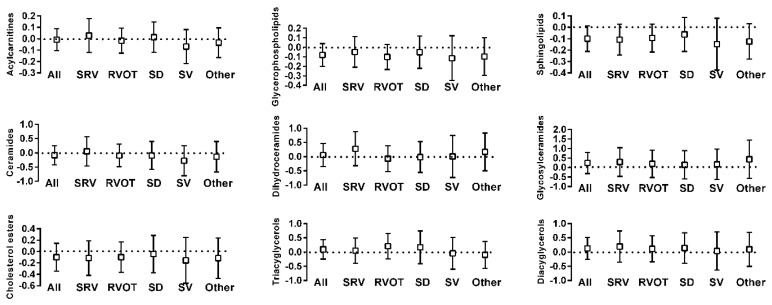
Comparison of concentration of lipid-derived metabolic compounds between patients with ACHD and controls (box is the normalized differences along with 95% confidence interval: Dotted line at x = 0 is the concentration in control subjects). Y-axis values indicate fold concentration difference between ACHD and controls for the indicated variable. x-axis abbreviations: ALL—all diagnoses; OTH—other diagnoses; SRV—transposition of the great arteries; RVOT—Tetralogy of Fallot/right ventricular outflow stenosis; SD—septal defect; SV—single ventricle.

**Figure 3 metabolites-11-00525-f003:**
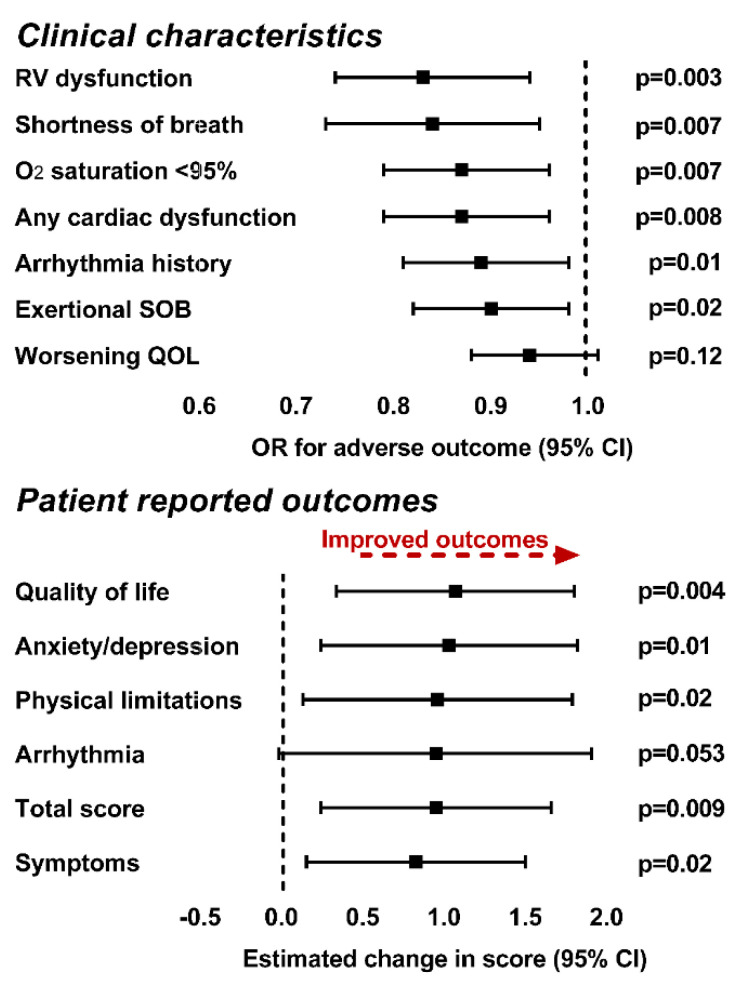
Association between patient score on PC4, clinical characteristics (top graph) and patient reported outcomes (bottom graph) in multivariable regression models. We found that higher plasma concentrations of the metabolites in PC4 were associated with both a decreased probability of adverse clinical characteristics and improved patient reported health status in all domains other than arrhythmia. Models for clinical outcomes (other than worsening quality of life) are adjusted for clinical covariates. Worsening quality of life is adjusted for age and TGA, anxiety/depression score, quality of life score, symptoms and total score are adjusted for the presence of non-cardiac diagnoses and the symptoms score is further adjusted for the sex of the participant. Abbreviations: CI—confidence interval; O2—oxygen; OR—odds ratio; QOL—quality of life; SOB—shortness of breath.

**Table 1 metabolites-11-00525-t001:** Patient characteristics (N = 99).

Characteristics.	N	Value
Sex (male)	99	50 (51%)
Diagnosis	99	
Transposition of the great arteries (SRV)		21 (21%)
TOF/DORV, RVOT stenosis (RVOT)		34 (35%)
Septal defects (SD)		20 (20%)
Single ventricle/Ebstein’s (SV)		9 (9%)
Other diagnoses		15 (15%)
Non-cardiac diagnosis	96	71 (74%)
Caucasian race		75 (76%)
Education	89	
High school or less		36 (40%)
College		22 (25%)
Post-college		31 (35%)
Smoking	98	
Never		64 (65%)
Past smoker		11 (11%)
Current smoker		23 (24%)
Hypertension	99	35 (35%)
Diabetes	99	14 (14%)
Dyslipidemia	98	18 (18%)
History of arrhythmia	99	45 (45%)
Previous vascular surgery	92	10 (11%)
Previous non-cardiac surgery	98	59 (60%)

Abbreviations: N—number of patients; TOF—Tetralogy of Fallot; DORV—double outlet right ventricle; RVOT—right ventricular outflow tract. Age changes between assessments and so is reflected in Table 2 with other time-dependent variables.

**Table 2 metabolites-11-00525-t002:** Significantly impacted pathways stratified by category of ACHD diagnoses (*** FDR *p* < 0.01, ** FDR 0.01 < *p* < 0.05, * FDR 0.05 < *p* < 0.10, no * FDR > 0.10).

Significantly Impacted Pathways	
**All congenital heart disease**	**Septal**
Aminoacyl-tRNA biosynthesis ***	Aminoacyl-tRNA biosynthesis *
Histidine metabolism ***	Histidine metabolism *
Arginine biosynthesis **	Arginine and proline metabolism
Arginine and proline metabolism *	Arginine biosynthesis
Phenylalanine, tyrosine and tryptophan biosynthesis *	beta-Alanine metabolism
Galactose metabolism	
Alanine, aspartate and glutamate metabolism	**Single ventricle**
D-Glutamine and D-glutamate metabolism	Taurine and hypotaurine metabolism
Taurine and hypotaurine metabolism	Sphingolipid metabolism
beta-Alanine metabolism	Linoleic acid metabolism
Phenylalanine metabolism	Cysteine and methionine metabolism
**Systemic right ventricle**	**Other**
Alanine, aspartate and glutamate metabolism ***	Histidine metabolism **
Histidine metabolism ***	Aminoacyl-tRNA biosynthesis *
Aminoacyl-tRNA biosynthesis **	Pantothenate and CoA biosynthesis
Arginine biosynthesis **	beta-Alanine metabolism
Arginine and proline metabolism **	Alanine, aspartate and glutamate metabolism
D-Glutamine and D-glutamate metabolism **	Arginine biosynthesis
beta-Alanine metabolism **	Glycerophospholipid metabolism
Butanoate metabolism	
Sphingolipid metabolism	
**TOF/DORV, RVOT stenosis**	
Arginine biosynthesis ***	
Aminoacyl-tRNA biosynthesis ***	
Histidine metabolism *	
D-Glutamine and D-glutamate metabolism *	
Arginine and proline metabolism *	
Taurine and hypotaurine metabolism	
Phenylalanine metabolism	
Alanine, aspartate and glutamate metabolism	
Pantothenate and CoA biosynthesis	

Abbreviations: FDR—false discovery rate; TOF—Tetralogy of Fallot; DORV—double outlet right ventricle; RVOT—right ventricular outflow tract.

**Table 3 metabolites-11-00525-t003:** Metabolic pathways represented in the first four components identified with principal component analysis (*** FDR *p* < 0.01, ** FDR 0.01 < *p* < 0.05, * FDR 0.05 < *p* < 0.10, no * FDR > 0.10).

Significantly Impacted Pathways	
**PC1**	**PC2**
Glycerolipid metabolism	Sphingolipid metabolism *
	Linoleic acid metabolism
**PC3**	alpha-Linolenic acid metabolism
Glycine, serine and threonine metabolism	
Glycerophospholipid metabolism	**PC4**
	Arginine biosynthesis *
	Phenylalanine metabolism
	Phenylalanine, tyrosine and tryptophan biosynthesis
	Galactose metabolism
	Butanoate metabolism
	D-Glutamine and D-glutamate metabolism
	Taurine and hypotaurine metabolism
	Arginine and proline metabolism

**Table 4 metabolites-11-00525-t004:** Multivariable models. Association between principal components loads, individual metabolites with high reliability in bootstrap resampling and adverse cardiac outcomes. Left side—regression models built with PC loads and clinical covariates only, reported are the variables included in the final regression model. Right side—regression models built with PC loads and clinical covariates (if statistically significant) and the highly reliable metabolites.

	PC Only			PC + Highly Reliable Metabolites		
Outcomes	PC Loads	Clinical Covariates	QIC (PC)	PC Loads	Significant Metabolites	QIC (mets)
O_2_ saturation < 95%	4	SV, SD	95.6		ADMA/Arg, serotonin/tryptophan, GCDCA, Betaine	84.6
Cardiac dysfunction	4	None	119.6	2,5	Citrulline, Cystine, 1-Methylhistidine	103.4
RV dysfunction	4	SRV, dyslipidemia	79.9	4	DHEAS	76.4
Arrhythmia history	2,4	SRV, SV, age	179.3	2	TCA, Orn/Arg	175.4
Shortness of breath	1,4	None	143.0		Ornithine, SDMA/Arg, Homo-L-arginine	120.9

ADMA—asymmetric dimethyl arginine; Arg—arginine; DHEAS—dehydroepiandrosterone; GCDCA—glycochenodeoxycholic acid; Orn—ornithine; SDMA—symmetric dimethyl arginine; TCA—Taurocholic acid.

## Data Availability

Data in this study will be provided to researchers upon written request with appropriate IRB approval.

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
