# Peer review of "Metabolomic Profiling of Adults with Congenital Heart Disease"

_metabolites, 2021, doi:10.3390/metabo11080525_

Round 1
Reviewer 1 Report
The manuscript titled " Metabolomic profiling of adults with congenital heart disease" is a very interesting article describing the role of some clinical risk factors in in patients with congenital heart diseases. Methods are properly reported and references are updated and of good quality. The introduction should be improved and a paragraph on the molecular pathwys in congenital heart disease should be added. Particular attention should be made on the role of cardiac microenvironment in cardiovascular events ( you can cite doi: 10.1097/FJC.0000000000001026. ). A small discussion on the lifestyle modifications aimed to improve quality of life in these patients should be made ( i.e prevention of hyperglycemia or the use of nutraceuticals ; you can cite doi: 10.3390/ijms21207802 and doi: 10.3390/antiox9121182. ). The manuscript will be acccepted after these modifications.
Author Response
English has been revised by a native speaker.

Reviewer 2 Report
The paper is good. Revise english. Best regards
Author Response

(The authors gave the same response as above.)

Reviewer 3 Report
The paper aims to generate hypotheses from the metabolomics profiling of blood from adults with congenital heart disease, with the aim of, as written by the authors, discovering “biomarkers for disease activity and suggest etiologically related pathways as possible targets for disease-modifying intervention”. This was performed by performing mass spec-based targeted metabolomics analysis and quantifying concentrations of metabolites of specific pathways.
Comments:
- The introduction is well written, highlighting the limitations of complex genetic/environmental factors behind these heterogeneous group of patients, thus using metabolomics to look for “unifying” metabolic phenotypes.
- On patient demographics: I understand the rarity of the heart condition, limiting the number of subjects that can be recruited. However in this study there are at least 4 heart related diagnoses, each group having between 9 to 34 patients. I am concerned about the validity of the statistics, especially as stated in the result section, that the first 4 principal component only explained 48% of the variance. In addition to the small population, there are also a number or clinical co-variates to consider. Would the authors please comment on the statistics used, and power of the study. I cannot comment further on the statistics as this is not within my field, but a clarifying statement will be useful for non-statistics trained readers in allaying concerns of the small population.
- Can PCA plot be added to the figures?
- Figure 1 and 2: The figures are comparing concentrations, can the authors please clarify on the unit of concentration (missing unit on x axis).
- Figure 2: Lipid derived metabolites were seen to be analysed as a total of individual species. It is useful to investigate the changes of the individual species, especially that of acylcarnitines, intermediates of the fatty acid oxidation, which are sensitive to oxygen levels. Specific acylcarnitine (and other lipid) changes have also been reported in numerous articles studying cardiovascular diseases, each having their own implications. It is thus worthwhile to study this in more depth.
- Limitations: I am concerned about the study using non-fasted blood and without regards for diurnal variation. While a study was cited to support the use of non-fasted blood, a check on the article showed limited conditions such as small sample size, and short fasting period with the longest being “3 or more” hours.
Author Response

(The authors gave the same response as above.)

Reviewer 4 Report
Thank you for the opportunity to review this body of work on metabolomic features of congenital heart disease in adults. I have some minor modifications and suggestions for the authors' consideration.
- You need a statistical comparison of available demographic data between the control and disease cohorts (e.g. age and gender, for sure)
- For Figure 1, can you be more clear on the criteria by which the metabolites of interest were identified. Please use box and whisker plots that include the individual data points so the reviewer can see the variation in the dataset. Also, please include statistical comparisons of the groups.
- Also for Figure 1, based on your methods it would appear you have used a fully quantitative Biocrates method to measure your metabolites of interest. Why are you using log-normalized differences in metabolites as your output? It seems that a presentation of actual metabolite concentrations would be more informative.
- For the PCA analysis, could you provide the PCA plot so the readers/reviewers can see how well the groups separate?
- For the PCA, it is also interesting that the majority of variance was accounted for by lipids and these did not show up as major discriminating metabolites between the healthy vs disease group.
- Please upload your metabolomics data to an opensource repository and reference in your manuscript (apologies if you did and I didn't see it).
Author Response

(The authors gave the same response as above.)
